# ZERO-SHOT IMAGE RESTORATION VIA DIFFUSION INVERSION

## ABSTRACT

Recently, various methods have been proposed to solve Image Restoration (IR) tasks using a pre-trained diffusion models leading to state-of-the-art performance. A common characteristic among these approaches is that they alter the diffusion sampling process in order to satisfy the consistency with the corrupted input image. However, this choice has recently been shown to be sub-optimal and may cause the generated image to deviate from the data manifold. We propose to address this limitation through a novel IR method that not only leverages the power of diffusion but also guarantees that the sample generation path always lies on the data manifold. One choice that satisfies this requirement is not to modify the reverse sampling , *i.e.*, not to alter all the intermediate latents, once an initial noise is sampled. This is ultimately equivalent to casting the IR task as an optimization problem in the space of the diffusion input noise. To mitigate the substantial computational cost associated with inverting a fully unrolled diffusion model, we leverage the inherent capability of these models to skip ahead in the forward diffusion process using arbitrary large time steps. We experimentally validate our method on several image restoration tasks. Our method SHRED achieves state of the art results on multiple zero-shot IR benchmarks especially in terms of image quality quantified using FID.

## 1 INTRODUCTION

Recent advances in the field of generative learning due to diffusion models (Ho et al., 2020; Song & Ermon, 2019) and better architecture choices (Karras et al., 2021; Zhang et al., 2022) have led to models capable of generating detailed high-resolution images. In addition to their data generation capabilities, these models also provide a smooth representation of the learned data distribution, which can be used for various applications, such as image restoration. Indeed, different approaches (Kawar et al., 2022; Lugmayr et al., 2022; Wang et al., 2023) have emerged to solve inverse problems using pretrained diffusion models. Those approaches start from a random noise vector as the diffusion input and recursively alter the diffusion sampling process in order to guarantee the data fidelity. After each diffusion reverse step, a projection-based measurement consistency step is added to ensure the consistency with the corrupted image. Chung et al. (2022b) shows that this procedure may cause the generated image to step outside the data manifold during the iterative denoising process.

In this work, we devise a new zero-shot image restoration method that we call in short SHRED (zero-SHot image REstoration via Diffusion inversion) which, by construction, ensures that the sample generation path always lies on the data manifold. To this end, we opt for a simple yet effective strategy: we do not alter the reverse sampling , *i.e.*, all the intermediate latents, once an initial noise is sampled. More specifically, SHRED use a pre-trained Denoising Diffusion Implicit Models (DDIM) (Song et al., 2020) and exploits the existing deterministic correspondence between noise and images in DDIMs by casting the inverse restoration problem as a latent estimation problem, *i.e.*, where the latent variable is the input noise to the diffusion model.Given a degraded image, the pre-trained generative model allows us to retrieve the optimal initial noise that, when applying the reverse diffusion sampling, generates the closest clean image under the learnt image distribution. Since SHRED is an optimization method that works by iteratively optimizing the initial noise, multiple forward passes of the diffusion process are required which can be computationally expensive. A single forward pass of a standard diffusion model (with 1000 steps) takes in the order of minutes on a typical modern GPU. Thus, a direct implementation of the optimization procedure in the case of the Diffusion model as a black box would be computationally demanding and ultimately impractical.

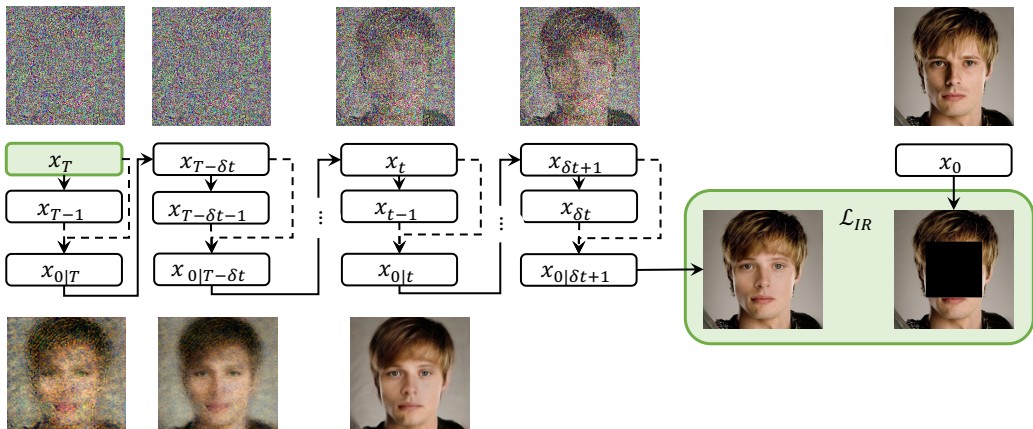

Figure 1: Image inpainting with SHRED. We show the flow during **one** iteration of our optimization scheme of initial noise $x_T$. The dashed and single line arrows denote the reverse and forward diffusion processes respectively. We optimize the inpainting error $\mathcal{L}_{IR}$ **only** with respect to the initial noise $x_T$. In our notation, $x_{0|t}$ is our intermediate estimate of $x_0$ given $x_t$. The top and bottom row images show $x_t$ and $x_{0|t}$ for different values of $t$ respectively. The reconstruction of $x_{0|t}$ allows us to skip time steps and make the generation process and the gradient back-propagation much more efficient.

To mitigate the substantial computational cost associated with inverting a fully unrolled diffusion model, we leverage the inherent capability of these models to skip ahead in the forward diffusion process using arbitrary large time steps. Thanks to the non-Markovian marginal distributions used in DDIM, our method introduces a hyperparameter $\delta t$ that can be used to prioritize either image quality or computational cost. We show that by optimizing for the best initial noise that corresponds to the unknown restored image, our method is able to achieve competitive image performance across a wide range of image restoration tasks both in the blind and non-blind setting. Our main contributions can be summarized as:

1. We are the first to cast the IR task as a latent optimization problem in the context of diffusion models where only the initial noise is optimized. This design choice does not alter the intermediate diffusion latents and thus provides more guarantees that the generated images lies in the in-distribution manifold.

2. We leverage the capabilty of DDIM to skip ahead in the forward diffusion process by arbitrary time steps and propose an efficient and practical diffusion model inversion in the context of image restoration. We note that in our method, DDIM is neither fine-tuned nor retrained.

3. We achieve competitive results on both CelebA and ImageNet for different IR tasks including image inpainting, super-resolution, compressed sensing and blind deconvolution.

## 2 RELATED WORK

**Zero-shot methods** Recently, methods that can be repurposed to different inverse problems have been called *zero-shot*. Song & Ermon (2019) proposes a method based on guiding the reverse diffusion process with the unmasked region to solve inpainting in a zero-shot manner. Song et al. (2021) proposes the use of gradient guidance to solve inverse problems in the context of medical imaging. Choi et al. (2021) applies low-frequency guidance from a reference image to solve super-resolution. RePaint (Lugmayr et al., 2022) solves the inpainting problem by guiding the diffusion process with the unmasked region. To condition the generation process, the reverse diffusion iterations are altered by sampling the unmasked regions using the given image information. DDRM (Kawar et al., 2022) proposes an inverse problem solver based on posterior sampling by introducing a variational inference objective for learning the posterior distribution of the inverse problem.

DDNM (Wang et al., 2023) proposes a zero-shot framework for IR tasks based on the range-null space decomposition. The method works by refining only the null-space contents during the reverse diffusion process,to satisfy both data consistency and realness. Fei et al. (2023) leverage DDPM and solve inverse problems via hierarchical guidance and a patch-based method. Chung et al. (2022b) argues that relying on an iterative procedure consisting of reverse diffusion steps and a projection-based consistency step runs the risk of stepping outside of the data manifold, a risk they mitigate using an additional correction term.

**Methods based on GAN inversion** Different approaches Pan et al. (2021); Yang et al. (2021); Yu et al. (2022) use a pre-trained GAN inversion to solve image restoration. Pan et al. (2021) use GAN as an image prior and estimates the latent that when fed to GAN will generates the clean image through optimization. The authors shows that this procedure alone does not achieve satisfactory results and propose to finetune the GAN while optimizing for the GAN latent. Yu et al. (2022) opt for encoder-based inversion to solve image inpainting. An encoder is trained to projects corrupted images into a latent space with a pre-modulation for learning more discriminative representation to solve the inpainting task. We note that SHRED is separate enough to the GAN inversion literature. In fact, GAN and diffusion models are fundamentally different. Thus, their inversion through optimization requires different techniques. For example, one of the main differences is that GANs generate samples in "one forward pass", while Diffusion models require multiple iterative steps. Moreover, all IR methods based on GAN inversion, either train an auxiliary network or fine-tune the pre-trained GAN network. In our method, the diffusion model is neither fine-tuned nor trained.

## 3 BACKGROUND

### 3.1 DENOISING DIFFUSION PROBABILISTIC MODELS

Denoising Diffusion Probabilistic Models (DDPM) (Ho et al., 2020) leverage diffusion processes in order to generate high quality image samples. The aim is to reverse the forward diffusion process that maps images to noise, either by relying on a stochastic iterative denoising process or by learning the explicit dynamics of the reverse process, *e.g.*, through an ODE (Song et al., 2020). More precisely the forward diffusion process maps an image $x_0 \sim p(x)$ to a zero-mean Gaussian $x_T \sim \mathcal{N}(0, \mathbf{I})$ by generating intermediate images $x_t$ for $t \in (0, T]$ which are progressively noisier versions of $x_0$. DDPM (Ho et al., 2020) adopts a Markovian diffusion process, where $x_t$ only depends on $x_{t-1}$. Given a non-increasing sequence $\alpha_{1:T} \in (0, 1]$, the joint and marginal distributions of the forward diffusion process are described by

$$q(x_{1:T}|x_0) = \prod_{t=1}^{T} q(x_t|x_{t-1}), \text{ where } q(x_t|x_{t-1}) = \mathcal{N}\left(\sqrt{\alpha_t}x_{t-1}, \left(1 - \alpha_t\right)\mathbf{I}\right), \quad (1)$$

which implies that we can sample $x_t$ simply by conditioning on $x_0$ with $\bar{\alpha}_t = \prod_{s \leq t} \alpha_s$ via

$$q(x_t|x_0) = \mathcal{N}\left(\sqrt{\bar{\alpha}_t}x_0, (1 - \bar{\alpha}_t)\mathbf{I}\right). \quad (2)$$

To invert the forward process, one can train a model $\epsilon_\theta$ to minimize the objective

$$\min_\theta \ \mathbb{E}_{t \sim \mathcal{U}(0,1); x_0 \sim q(x); \epsilon \sim \mathcal{N}(0,\mathbf{I})}\left[\|\epsilon - \epsilon_\theta(\sqrt{\bar{\alpha}_t}x_0 + \sqrt{1 - \bar{\alpha}_t}\epsilon, t)\|^2\right]. \quad (3)$$

Image samples $\hat{x}_0$ given the initial noise $x_T$, are obtained by iterating for $t \in [1, T]$ the denoising update

$$x_{t-1} = 1/\sqrt{\alpha_t}(x_t - \epsilon_\theta(x_t, t) \times (1-\alpha_t)/\sqrt{1-\bar{\alpha}_t}) + \sigma_t z, \quad (4)$$

with $z \sim \mathcal{N}(0, \mathbf{I})$, and $\sigma_t^2 = \frac{1-\bar{\alpha}_{t-1}}{1-\bar{\alpha}_t}(1 - \alpha_t)$.

Song et al. (2020) point out that the quality of generated images highly depends on the total number of denoising steps $T$. Thus, the inference loop using Eq. (4) becomes computationally expensive. To reduce the computational cost, they propose a Denoising Diffusion Implicit Model (DDIM) (Song et al., 2020), which foregoes the Markovian assumption in favor of a diffusion process $q(x_{1:T}|x_0) = q(x_T|x_0) \prod_{t=2}^{T} q(x_{t-1}|x_t, x_0)$ where $q(x_T|x_0) = \mathcal{N}\left(\sqrt{\bar{\alpha}_T}x_0, (1 - \bar{\alpha}_T)\mathbf{I}\right)$ and

$$q(x_{t-1}|x_t, x_0) = \mathcal{N}\left(\sqrt{\bar{\alpha}_{t-1}}x_0 + \sqrt{1 - \bar{\alpha}_{t-1} - \sigma_t^2} . (x_t - \sqrt{\bar{\alpha}_t}x_0)/\sqrt{1-\bar{\alpha}_t}, \sigma_t^2\mathbf{I}\right). \quad (5)$$

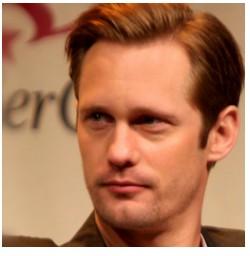 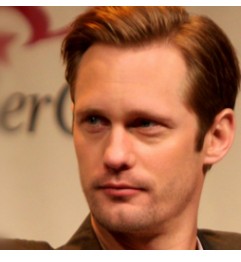 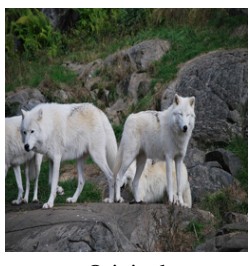 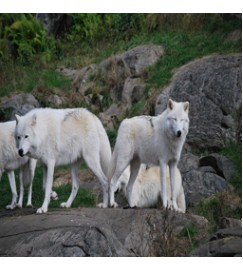

| Original | Reconstructed (PSNR=$44.10 \pm 1.06$) | Original | Reconstructed (PSNR=$42.43 \pm 0.67$) |

Figure 2: Inversion results using our optimization scheme on samples from both CelebA and ImageNet validation datasets. The PSNR mean and standard deviation are computed over 10 runs.

When $\sigma_t = 0$ for all $t$, the diffusion process is fully deterministic. This means that when we start from the same noisy sample $x_T$ we obtain the same generated image sample. Given $x_t$, one can first predict the denoised observation $\hat{x}_0$, which is a prediction of $x_0$ given $x_t$

$$\hat{x}_0 = \left(x_t - \sqrt{1-\bar{\alpha}_t}\epsilon_\theta(x_t, t)\right)\big/\sqrt{\bar{\alpha}_t}. \qquad (6)$$

then we can predict $x_{t-1}$ from $x_t$ and $\hat{x}_0$ using Eq. (5) by setting $\sigma_t = 0$

$$x_{t-1} = \sqrt{\bar{\alpha}_{t-1}}\hat{x}_0 + \sqrt{1-\bar{\alpha}_{t-1}}\hat{\omega} \qquad (7)$$

with $\hat{\omega} = \frac{x_t - \sqrt{\bar{\alpha}_t}\hat{x}_0}{\sqrt{1-\bar{\alpha}_t}}$ a direction pointing to $x_t$.

Song et al. (2020) shows that this formulation allows DDIM to use fewer time steps at inference by directly predicting $x_{t-k}$ with $k > 1$ which results in a more computationally efficient process using

$$x_{t-k} = \sqrt{\bar{\alpha}_{t-k}}\hat{x}_0 + \sqrt{1-\bar{\alpha}_{t-k}}\hat{\omega}. \qquad (8)$$

# 4 IMAGE RESTORATION VIA SHRED

In this section, we introduce the general setting of inverse problems that we aim to solve and how generative models can be used to address the ambiguities associated with this class of image restoration tasks. We then present our zero-shot image restoration algorithm SHRED.

## 4.1 AN UNBIASED FORMULATION

Image restoration (IR) can be cast as a Maximum a Posteriori (MAP) optimization problem (Zhang et al., 2020)

$$\hat{x} = \underset{x \in \mathbb{R}^{N_x \times M_x}}{\arg\max} \ \log p(y|x) + \log p(x), \qquad (9)$$

where $\hat{x} \in \mathbb{R}^{N_x \times M_x}$ is the restored image and $y \in \mathbb{R}^{N_y \times M_y}$ is the degraded image, and where $p(y|x)$ is the so-called likelihood and $p(x)$ the prior distribution. Linear inverse problems describe IR tasks with linear degradation operators $H : \mathbb{R}^{N_x \times M_x} \to \mathbb{R}^{N_y \times M_y}$ and often consider zero-mean additive noise $\eta$, such that the observed degraded image $y$ can be written as

$$y = H(x) + \eta. \qquad (10)$$

IR tasks such as image inpainting, denoising and super-resolution can all be described using specific choices of $H$ and $\eta$. For instance, if we vectorize the image $x$ and the output of the operator $H$, we can describe image inpainting via a pixel projection operator $H(x)_i = x_{k[i]}$, where $i = 1, \ldots, N_y \times M_y$, $k$ is a permutation of $N_x \times M_x > N_y \times M_y$ indices, and $\eta = 0$. The associated $H$ and $\eta$ for each image restoration task[1] considered in this paper are provided in the supplementary material. By

---

[1] A more accurate denomination of the operators used in the linear image restoration problems should be *affine* rather than linear. Nonetheless, we follow the terminology used in the literature for simplicity.

---

**Algorithm 1** SHRED: Image Restoration via DDIM inversion

---

**Require:** Degraded image $y$, step size $\delta t$, learning rate $\alpha$
**Ensure:** Restored image $\hat{x}_0$
1: Initialize $x_T$ with random from $\mathcal{N}(0, 1)$. If blind, initialize $H_\theta$.
2: **for** $k : 1 \rightarrow N$ **do**
3:      Initialize t = T
4:      **while** t > 0 **do**
5:          $\hat{x}_{0|t} = (x_t - \sqrt{1 - \bar{\alpha}_t}\epsilon_\theta(x_t, t))/\sqrt{\bar{\alpha}_t}$
6:          $x_{t-\delta t} = \sqrt{\bar{\alpha}_{t-\delta t}}\hat{x}_{0|t} + \sqrt{1 - \bar{\alpha}_{t-\delta t}} \cdot \frac{x_t - \sqrt{\bar{\alpha}_t}\hat{x}_{0|t}}{\sqrt{1-\bar{\alpha}_t}}$
7:          t = t - $\delta t$
8:      **end while**
9:      **if** BLIND **then**
10:          $x_T^{k+1} = x_T^k - \alpha\nabla_{x_T}\mathcal{L}_{IR}(x_T^k, H_{\theta^k})$
11:          $\theta^{k+1} = \theta^k - \alpha\nabla_\theta\mathcal{L}_{IR}(x_T^k, H_{\theta^k})$
12:      **else**
13:          $x_T^{k+1} = x_T^k - \alpha\nabla_{x_T}\mathcal{L}_{IR}(x_T^k)$
14:      **end if**
15: **end for**
16: Return the restored image $\hat{x}_0$

---

applying the desired observation model to the MAP formulation, one can aim to recover $\hat{x}$ given $y$ by using

$$\hat{x} = \underset{x \in \mathbb{R}^{N_x \times M_x}}{\arg\min} \|y - H(x)\| + \lambda\mathcal{R}(x), \qquad (11)$$

where $\|.\|$ is often a norm that depends on the nature of the noise $\eta$, *e.g.*, a zero-mean Gaussian noise $\eta$ induces a squared $L_2$ norm, $\mathcal{R}$ is a term corresponding to the prior, and $\lambda > 0$ is a coefficient that regulates the interplay between the likelihood and the prior.

In plug-and-play methods (Zhang et al., 2020) the prior $\mathcal{R}$ is obtained through a pre-training that aims at denoising the images $x$. However, using the prior $p(x)$ also captures the bias of the specific dataset used for pre-training. Since we do not know where the prior will be used, we argue that a bias could impact negatively the restoration process. Thus, we propose to employ a formulation that implicitly assumes a uniform prior on the domain $\Omega \subset \mathbb{R}^{N_x \times M_x}$ of valid reconstructions, *i.e.*, $\mathcal{R}(x) = -\log(p_{\mathcal{U}}(x)) = -\log(\text{const} \cdot \mathbf{1}_\Omega(x))$, where $\mathbf{1}_\Omega(x)$ is 1 for $x$ in the support $\Omega$ and 0 otherwise. This choice of $\mathcal{R}(x)$ is equivalent to considering an unbiased prior over the support of the data distribution and results in the following formulation

$$\hat{x} = \arg\min_{x \in \Omega} \|y - H(x)\|. \qquad (12)$$

To ensure that $x \in \Omega$, we parameterize $x$ via the initial noise of a pre-trained diffusion model. In the next section, we present our approach to infer the initial noise such that the generated image minimizes Eq. (12).

## 4.2 AN EFFICIENT DIFFUSION INVERSION

In this work, we adopt DDIM as our pre-trained generative process and we adopt a fully deterministic diffusion process ($\sigma_t = 0$). We propose a new iterative inversion process, which we summarize in Algorithm 1. We adopt an optimization-based iterative procedure to solve Eq. (12). In contrast to Kawar et al. (2022); Wang et al. (2023); Lugmayr et al. (2022), we explicitly use the correspondence between image samples $x_0$ and their associated latent vector $x_T$. SHRED starts from an initial noise instance $x_T^0 \sim \mathcal{N}(0, \mathbf{I})$ and at the end of each iteration $k$, we update $x_T^k$. At each iteration, we first predict $\hat{x}_{0|t}$ from $x_t$ using the pre-trained DDIM model $\epsilon_\theta$.

$$\hat{x}_{0|t} = \frac{x_t - \sqrt{1 - \bar{\alpha}_t}\epsilon_\theta(x_t, t)}{\sqrt{\bar{\alpha}_t}}. \qquad (13)$$

Rather than denoising $x_T$ iteratively for all the steps used in the pre-training, we make larger steps by using intermediate estimates of $\hat{x}_{0|t}$. We define a hyper-parameter $\delta t$ that controls the number of denoising steps and we can directly jump to estimate $x_{t-\delta t}$ from $\hat{x}_{0|t}$ and $x_t$ using

$$x_{t-\delta t} = \sqrt{\bar{\alpha}_{t-\delta t}}\hat{x}_{0|t} + \sqrt{1 - \bar{\alpha}_{t-\delta t}} \cdot \frac{x_t - \sqrt{\bar{\alpha}_t}\hat{x}_{0|t}}{\sqrt{1-\bar{\alpha}_t}}. \qquad (14)$$

Table 1: LPIPS/FID metrics for different degradation tasks on the CelebA dataset. The best and second best methods are in bold and underlined respectively.

| Method | Inpainting | SR ×8 | SR ×16 | SR ×32 | Blind Deconv | CS 1% | CS 5% |
|---|---|---|---|---|---|---|---|
| DIP Ulyanov et al. (2018) | 0.373/125.56 | 0.423/195.74 | 0.507/228.14 | 0.581/267.27 | - | - | - |
| SelfDeblur Ren et al. (2020) | - | - | - | - | 0.667/343.40 | - | - |
| Repaint Lugmayr et al. (2022) | 0.028/7.82 | - | - | - | - | 0.583/181.57 | 0.389/142.61 |
| DDRM Kawar et al. (2022) | 0.039/8.69 | 0.239/123.54 | 0.385/139.15 | 0.492/149.12 | - | 0.567/162.22 | 0.462/149.79 |
| DDNM Wang et al. (2023) | **0.018/6.09** | **0.299**/113.76 | 0.376/122.95 | 0.447/143.86 | - | 0.570/179.43 | **0.319**/105.06 |
| SHRED | 0.021/6.51 | 0.304/**78.84** | **0.325/85.82** | **0.416/110.75** | **0.362/90.28** | 0.558/**139.88** | 0.344/**85.42** |

Higher step size $\delta t$ allows us to favor speed while a lower one will favor fidelity. This iterative procedure results in an estimate of $x_0$ that is differentiable in $x_T$. We denote by $f$ the mapping between $x_T$ and $x_t$. The estimate $x_0 = f(x_T)$ is used to update $x_T$ such that it minimizes the image restoration loss. Our method supports both non-blind and blind image degradation.

**Non-blind case** In the non-blind case, $H$ is supposed to be known, the restoration loss becomes

$$\mathcal{L}_{IR}(x_T) = \|y - H(f(x_T))\| \tag{15}$$

Given an initial noise $x_T^0$, the optimization iteration is simply derived as the following

$$x_T^{k+1} = x_T^k - \alpha \nabla_{x_T} \mathcal{L}_{IR}(x_T^k) \tag{16}$$

where $\alpha$ is the learning rate.

**Blind case** In the blind case, $H$ is unknown and can be approximated by a parametric function $H_\theta$, the restoration loss becomes

$$\mathcal{L}_{IR}(x_T, H_\theta) = \|y - H_\theta(f(x_T))\| \tag{17}$$

Given an initial noise $x_T^0$ and $H_{\theta^0}$, the optimization iteration is simply derived as the following

$$x_T^{k+1} = x_T^k - \alpha \nabla_{x_T} \mathcal{L}_{IR}(x_T^k, H_{\theta^k}) \tag{18}$$

$$\theta^{k+1} = \theta^k - \alpha \nabla_\theta \mathcal{L}_{IR}(x_T^k, H_{\theta^k}) \tag{19}$$

where $\alpha$ is the learning rate.

One might wonder if the proposed fast inversion procedure suffers from local minima, such that different initial noise samples $x_T$ might result in different estimates of $x_0$. We show in Fig. 2 that our inversion method is stable and is insensitive to the initialization. We use images from both ImageNet and CelebA and perform denoising without noise (*i.e.*, $H$ is the identity function and $\eta = 0$). As can be observed, the proposed inversion method is stable.

## 5 EXPERIMENTS

### 5.1 IMAGE RESTORATION TASKS

In this work, we showcase our proposed method for different image restoration tasks. Specifically, we evaluate our method on 4 typical noise-free IR tasks, including image inpainting, ×4, ×8, ×16 and ×32 image super-resolution (SR), blind deconvolution, and compressed sensing (CS) with a 1% and 5% sampling rates. For our super-resolution experiments, we adopt a uniform downsampling. For the case of blind deconvolution, we use an anisotropic Gaussian kernel and we follow Ren et al. (2020) and use the same feed-forward network for the kernel estimation. Our compressed sensing experiments use a uniform pixel sampling strategy. Lastly, for our inpainting task, we consider a square centered mask. For each image restoration task, we use the same degradation operator for all methods. We evaluate our method both on ImageNet (Deng et al., 2009) and CelebA (Liu et al., 2018) at 256×256 resolution. We follow Chung et al. (2022a;b) and report the LPIPS Zhang et al. (2018) and FID Heusel et al. (2017) values obtained for each experiment.

Table 2: LPIPS/FID metrics for different degradation tasks on the ImageNet dataset. The best and second best methods are highlighted in bold and underlined respectively.

| Method | Inpainting | SR ×4 | SR ×8 | SR ×16 | Blind Deconv | CS 1% | CS 5% |
|---|---|---|---|---|---|---|---|
| DIP Ulyanov et al. (2018) | 0.175/75.12 | 0.369/114.83 | 0.544/267.31 | 0.738/376.35 | - | - | - |
| SelfDeblur Ren et al. (2020) | - | - | - | - | 0.719/368.94 | - | - |
| Repaint Lugmayr et al. (2022) | 0.068/7.85 | - | - | - | - | 0.769/338.31 | 0.623/292.17 |
| DDRM Kawar et al. (2022) | **0.042/5.32** | 0.302/75.34 | 0.478/195.82 | 0.578/277.86 | - | 0.784/345.39 | 0.607/295.52 |
| DDNM Wang et al. (2023) | 0.056/7.13 | **0.284**/70.88 | 0.450/183.64 | 0.582/281.15 | - | **0.700**/315.86 | **0.532**/276.63 |
| SHRED | 0.063/6.45 | 0.329/ **65.78** | **0.394/152.74** | **0.470/183.11** | **0.467/185.37** | 0.736/**295.12** | 0.642/**262.17** |

## 5.2 EXPERIMENTAL RESULTS

Our experimental results are summarized in Tables 1 and 2 for CelebA and ImageNet, respectively. We report the performance of SHRED as well as DIP (Ulyanov et al., 2018), SelfDeblur (Ren et al., 2020), RePaint (Lugmayr et al., 2022), DDRM (Kawar et al., 2022) and DDNM (Wang et al., 2023). We showcase the versatile nature of SHRED and its ability to restore degraded images with a higher level of details thanks to its explicit use of the correspondence property of the underlying pre-trained DDIM Song et al. (2020) model. We report the performance of super-resolution methods in Tables 1 and 2 for the ×8, ×16 and ×32 upscaling factors on CelebA and ×4, ×8 and ×16 on ImageNet respectively.

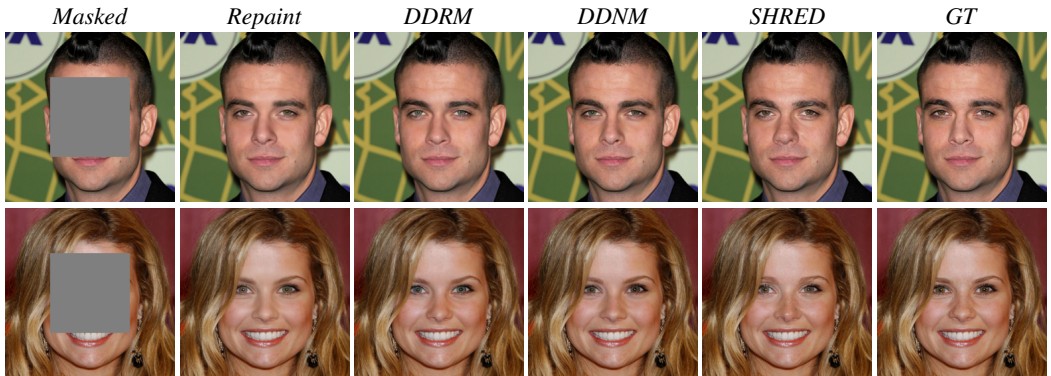

Figure 3: Visual results of inpaintings on CelebA.

For both datasets, we observe that our method is ranked first in both LPIPS and FID. We observe a similar pattern as in our super-resolution experiments for compressed sensing where SHRED generates more detailed images and achieves a better FID while being on par or slightly worse in terms of LPIPS.

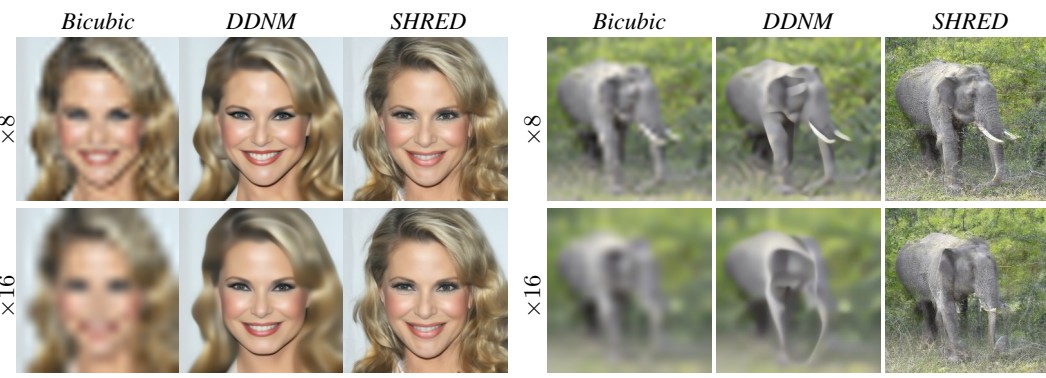

Figure 4: Visual results of super-resolution with different upsampling factors.

For the case of blind deconvolution, SHRED largely outperforms SelfDeblur (Ren et al., 2020) both in terms of FID and LPIPS which highlights its ability to also tackle blind restoration tasks through joint optimization. We illustrate this in Fig. 5 on two blurred images using a large blur kernel. Due to the significant blur, SelfDeblur is unable to restore the degraded image. SHRED on the other hand successfully deblurs the image especially around the main object in the image. Lastly, in the case of image inpainting SHRED is ranked second and third on CelebA and ImageNet respectively. On CelebA, SHRED achieves a comparable FID and LPIPS to DDNM while significantly outperforming other inpainting methods. We showcase the different considered methods in Fig. 3, where we observe that SHRED is capable of generating highly detailed and consistent features in the masked regions.

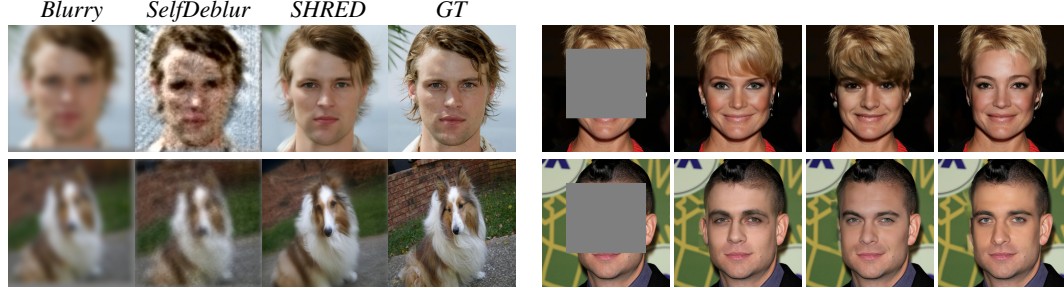

Figure 5: Visual results of blind deconvolution using a random Gaussian kernel with size $41 \times 41$.

Figure 6: Robustness of SHRED to the initial noise $x_T$.

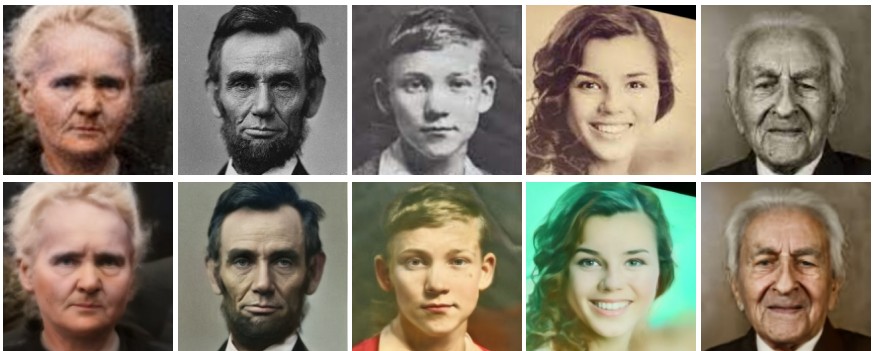

Figure 7: Restoring real-world photos using SHRED. Degraded images are collected from the internet

## 6 ABLATIONS STUDY

### 6.1 EFFECT OF DIFFERENT HYPER-PARAMETERS

We study the influence of hyper-parameters: the learning rate $\alpha$, the number of iterations $N$ and the step size $\delta t$, on SHRED performance. Table 4 shows the effect of varying the step size $\delta t$ on both FID and the computation time. $\delta t = 100$ gives the best trade-off between the speed and the visual quality. In Table 5, we report the FID values achieved by SHRED for different learning rates. SHRED acheives competitive FID with the different learning rates and this shows that it is robust enough to the choice of learning rate. Table 6 shows SHRED performance for different values of $N$ (the total number of iterations). SHRED needs around 100 iterations to achieve the best FID indicating that a limited number of iterations is sufficient to produce a decent reconstruction. We note that $N$ can be set automatically using the criteria $\mathcal{L}_{IR} < \epsilon$ . We empirically observe that tuning on a small set generalizes well in the zero-shot setting. For all IR tasks, $\delta t = 100$ and $\alpha = 0.003$ produces competitive performance.

| Method | Time[s] |
|---|---|
| DIP Ulyanov et al. (2018) | 520 |
| DDRM Kawar et al. (2022) | 13 |
| DDNM Wang et al. (2023) | 12 |
| *Naive Inversion* | 5750 |
| SHRED | 55 |

Table 3: Runtime (in seconds) comparison of different methods on CelebA per image on super-resolution. All experiments are conducted on a Geforce GTX 1080 Ti.

| Step size ($\delta t$) | 50 | 100 | 200 |
|---|---|---|---|
| FID | 6.38 | 6.51 | 8.32 |
| Time/Iteration (ms) | 2330 | 1150 | 576 |

Table 4: Effect of the step size $\delta t$ on the visual quality and the runtime of image inpainting on CelebA.

## 6.2 SHRED ROBUSTNESS

**Robustness to the initial noise**

We fix the degraded image and we test our method by varying the initial noise $x_T^0$. Figure6 shows that SHRED is robust to the input noise initialization and that for each time it can output a different plausible solution.

**Robustness to real-world degradations**

We apply our method to the task of old image enhancement where the images are real and collected from internet depicting complex and unknown degradation (noise, jpeg artifacts, blur..). The problem is framed as the mixture of image colorization and super-resolution. Figure 7 shows SHRED is able to enhance the image quality and that is robust to the different degradation.

## 6.3 COMPUTATIONAL COST

In Table 3, we report the runtime per image of the different compared methods. We denote by *Naive Inversion* a similar approach as SHRED that works by inverting the fully unrolled diffusion model. SHRED is $\times 100$ time faster than the *Naive Inversion*. Despite being an iterative method, SHRED has a reasonable runtime making it a practical zero-shot IR method. We note that the reported speed is using $\delta t = 100$ and a further speed boost is possible by opting for a higher value of $\delta t$ at a small reduction of the visual quality.

| $\alpha$ | $5 \times 10^{-4}$ | $10^{-3}$ | $2.5 \times 10^{-3}$ | $5 \times 10^{-3}$ |
|---|---|---|---|---|
| FID | 87.85 | 72.36 | 74.52 | 77.85 |

Table 5: Effect of the learning rate $\alpha$ on the visual quality of SR on CelebA.

| $N$ | 50 | 75 | 100 | 125 | 150 |
|---|---|---|---|---|---|
| FID | 78.03 | 75.63 | 74.31 | 73.85 | 74.42 |

Table 6: Effect of the number of iterations $N$ on the visual quality of SR on CelebA.

## 7 CONCLUSION

We have introduced SHRED, a zero-shot framework for solving IR tasks by using pre-trained diffusion generative models as learned priors. Our method exploits the deterministic correspondence between noise and images in DDIM by casting the inverse restoration problem as a latent estimation problem. We leverage the capabilty of DDIM to skip ahead in the forward diffusion process and provide an efficient diffusion inversion in the context of IR. SHRED is comprehensively utilized on various tasks such as super-resolution, inpainting, blind-deconvolution, and compressed sensing , demonstrating the capabilities of SHRED on unified image restoration.

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

## A  APPENDIX

### A.1  IMPLEMENTATION DETAILS

For evaluation, We choose a set of 100 random images from ImageNet 1K and CelebA 1K datasets with image size $256 \times 256$ for validation. For a fair comparison, we use the same pre-trained denoising models used in Wang et al. (2023) and apply them to all the diffusion-based methods. We use the Adam optimizer all IR tasks. The step size $\delta t$ is set to 100. All experiments are conducted on a Geforce GTX 1080 Ti.

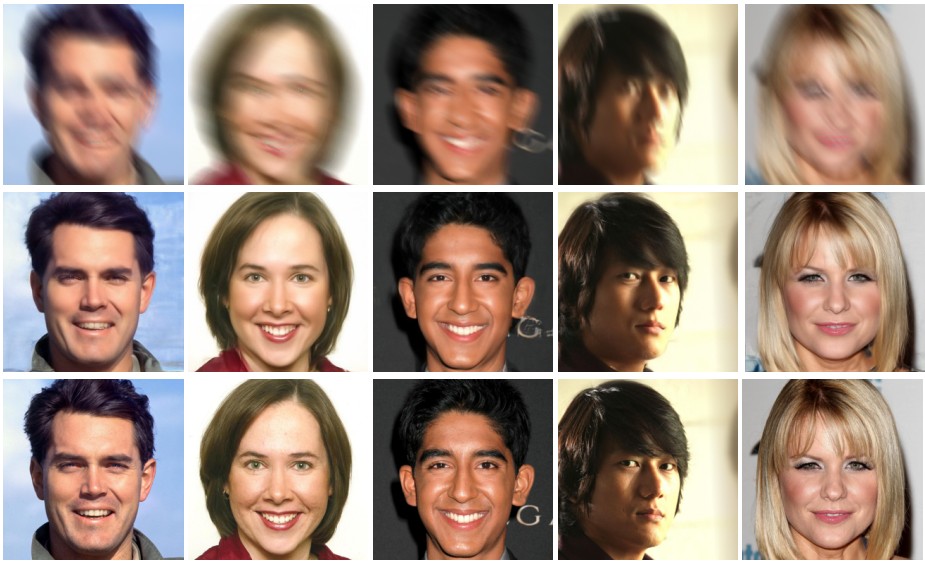

Figure 8: An application of SHRED on a non-linear inverse problem (non-uniform deblurring). The top row shows degraded images with non-uniform blur. The second row depicts the results of our method. The bottom row shows the ground truth.

### A.2  SUPER-RESOLUTION

For superresolution, we use $\times 4$, $\times 8$, and $\times 16$ uniform downsampling factors for ImageNet and $\times 8$, $\times 16$, and $\times 32$ uniform downsampling factors for CelebA. During training, we use Adam as our optimizer with a learning rate of $0.003$ and $50$ iterations. We also add comparisons with both OID Chen et al. (2020b) and Chen et al. Chen et al. (2020a) . Additional visual results are depicted in Fig. 11, Fig. 12 and Fig. 13. While other methods tend to generate over-smoothed images especially at higher downsampling factors, our method (SHRED) outputs sharper and more realistic images. For instance, SHRED is able to generate detailed grass and hair textures in Fig. 11 and Fig. 12 especially at $\times 16$ upscaling where other methods fail.

### A.3 BLIND DECONVOLUTION

For Blind Deconvolution, we apply a random anisotropic Gaussian kernel of size 41x41. During training, we use Adam as our optimizer with a learning rate of $0.003$ and 150 iterations. Visual results are shown in Fig. 16 and Fig. 17. Despite the heavy blur, SHRED is able to better deblur the main object present in the considered image compared to other methods.

### A.4 COMPRESSED SENSING (CS)

For Compressed Sensing, we randomly subsample $1\%$ and $5\%$ of the total available pixels. During training, we use Adam as our optimizer with a learning rate of $0.01$ and 150 iterations. More visual results are depicted in Fig. 14 and Fig. 15. SHRED performs better than the other methods at preserving the identity of the original image especially at the higher subsampling rates. Indeed, we see in the second and forth rows of Fig. 14 that other method hallucinate new faces instead of reconstructing the correct one.

### A.5 INPAINTING

For Inpainting, we apply a square mask for CelebA and a text mask for ImageNet. During training, we use Adam as our optimizer with a learning rate of $0.01$ and 200 iterations. We show additional results with different masks in Fig. 9 and Fig. 10.

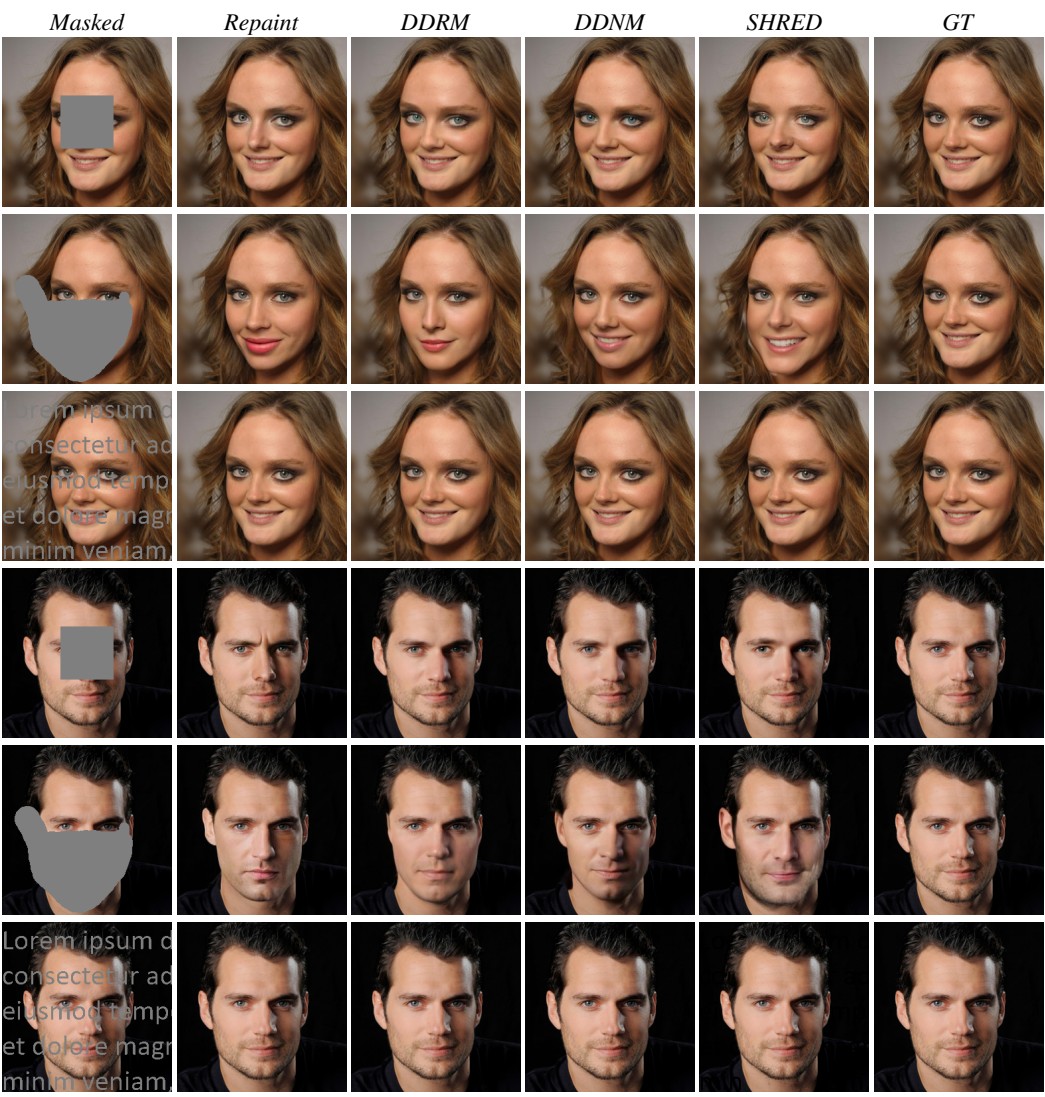

Figure 9: Visual results of inpaintings on CelebA.

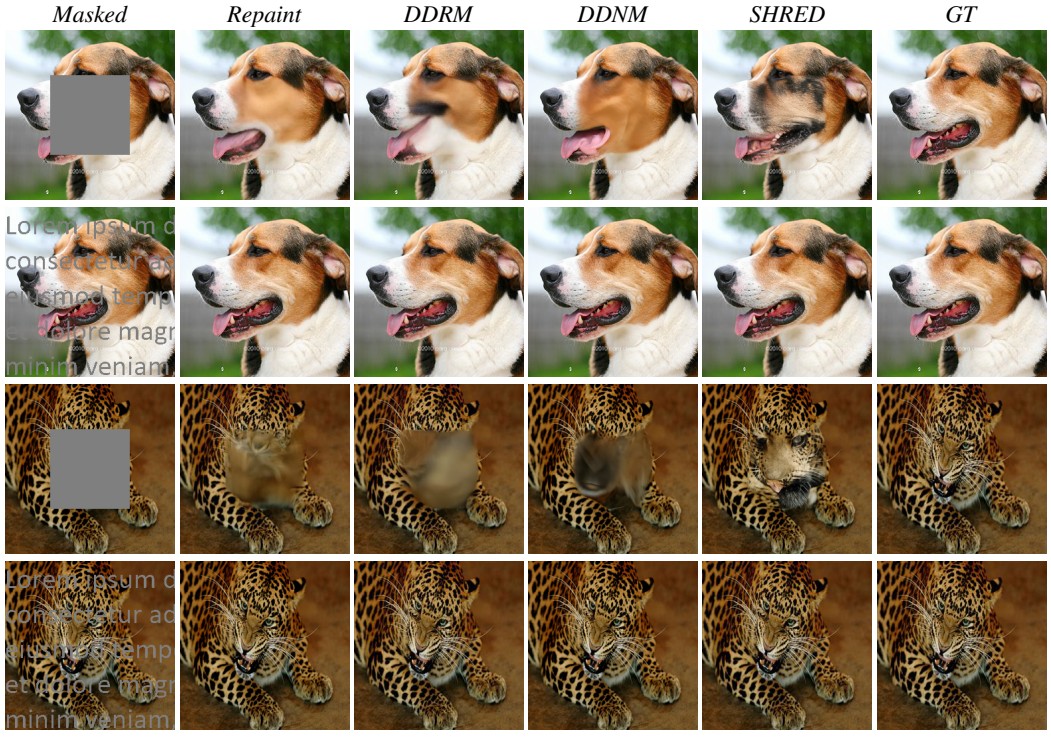

Figure 10: Visual results of inpaintings on ImageNet.

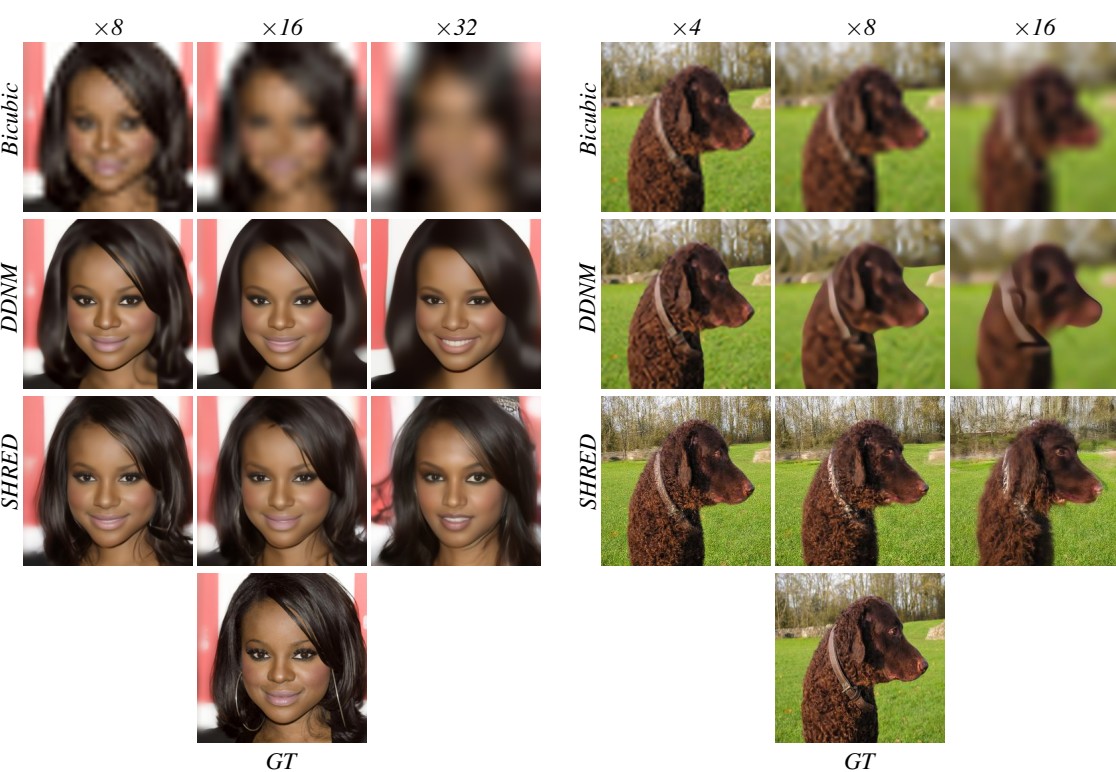

Figure 11: Visual results of super-resolution with different upsampling factors.

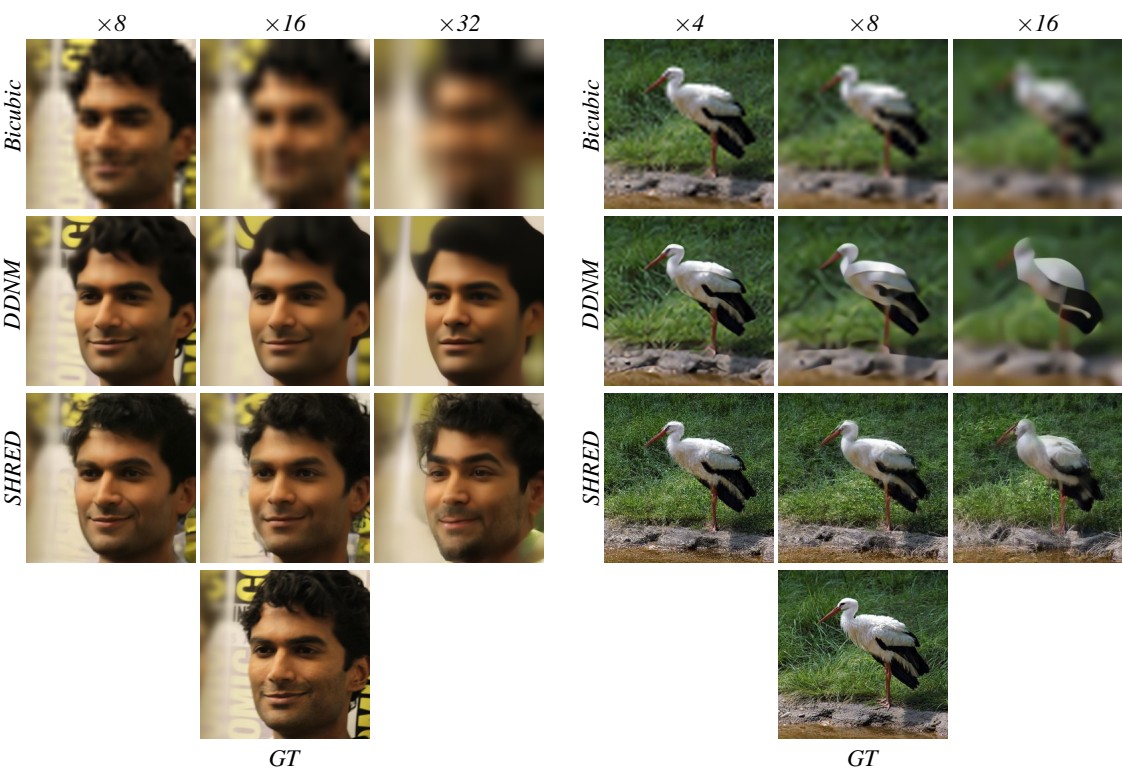

Figure 12: Visual results of super-resolution with different upsampling factors.

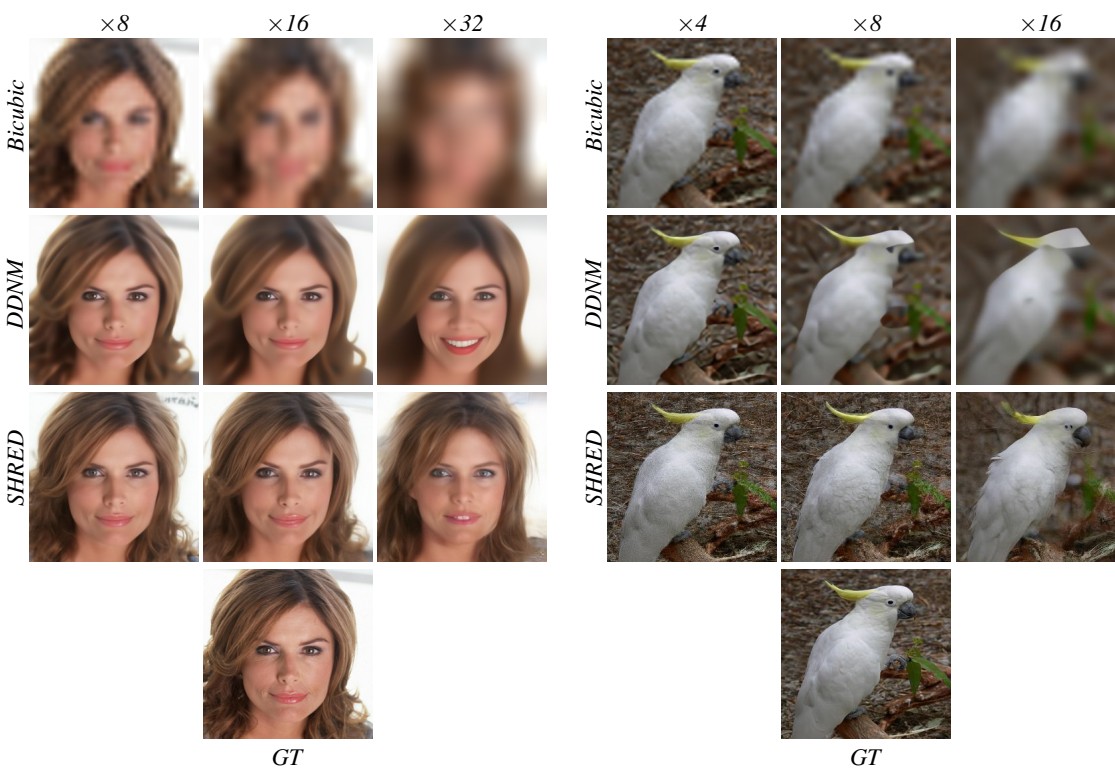

Figure 13: Visual results of super-resolution with different upsampling factors.

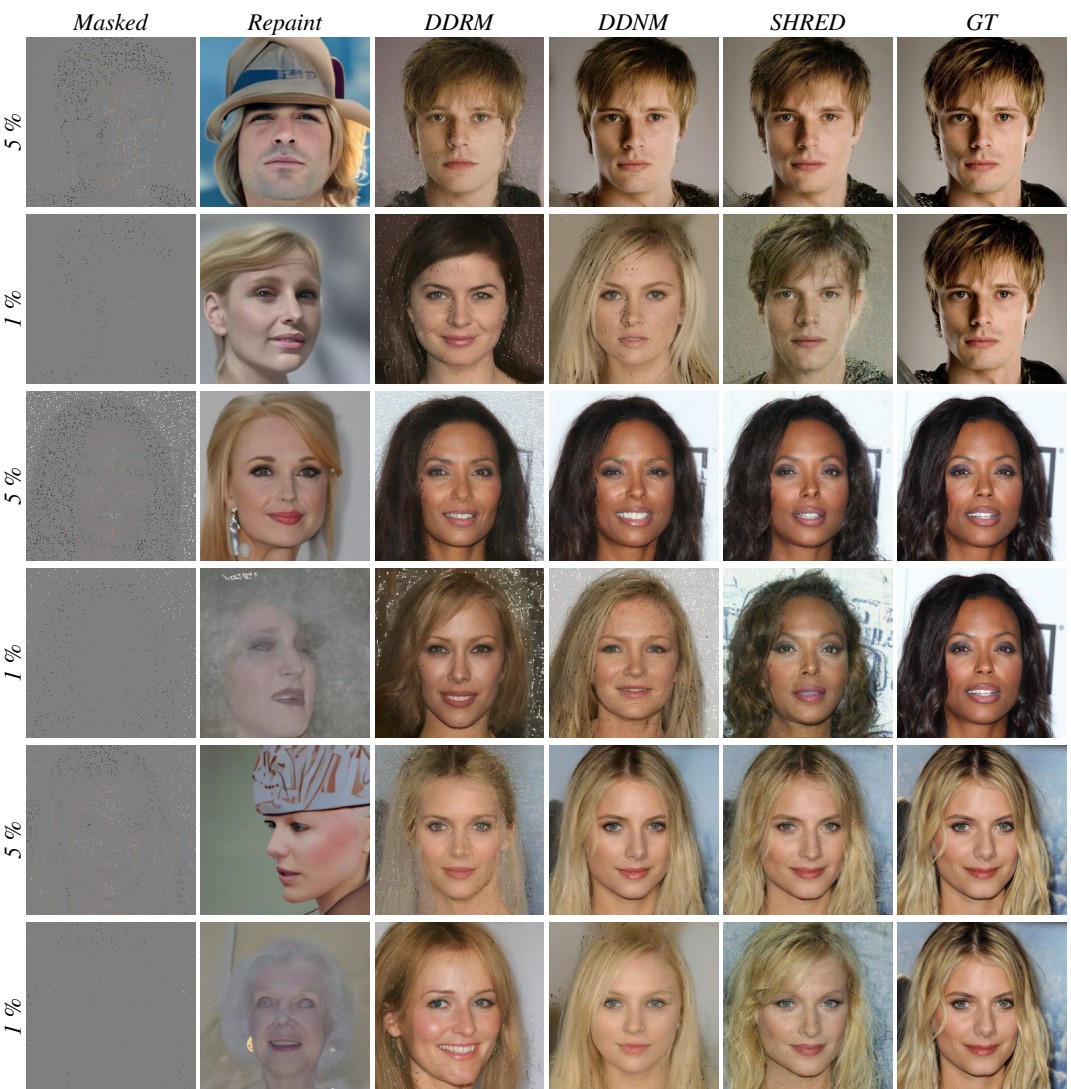

Figure 14: Visual results of CS on CelebA.

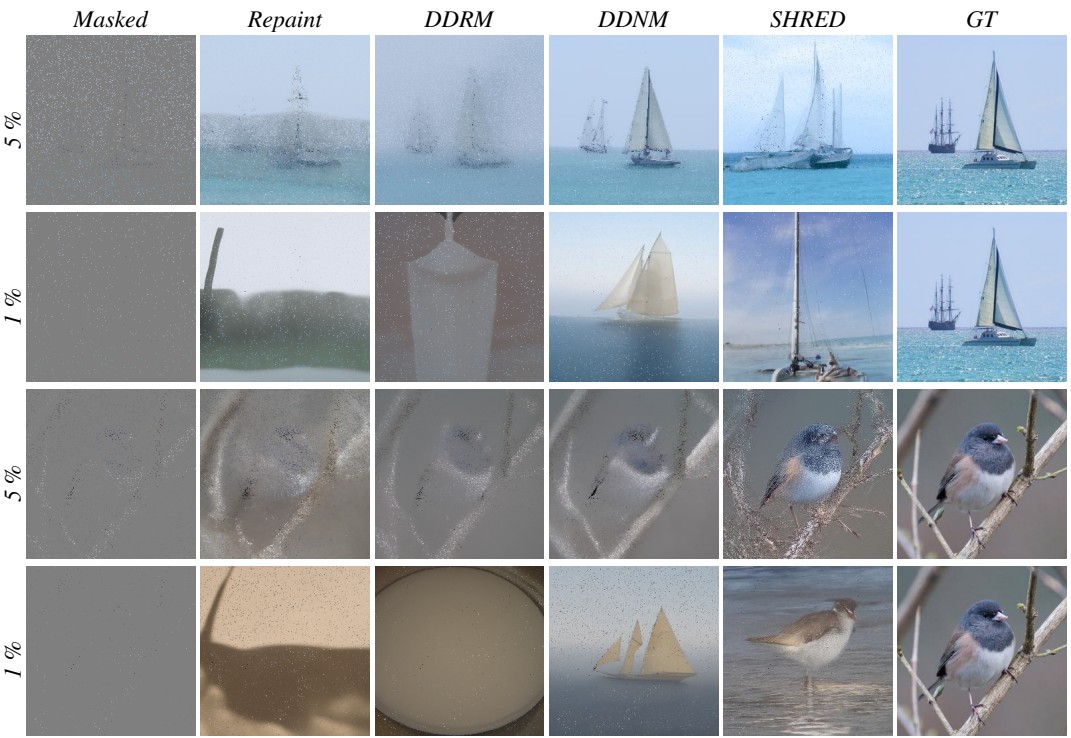

Figure 15: Visual results of CS on ImageNet.

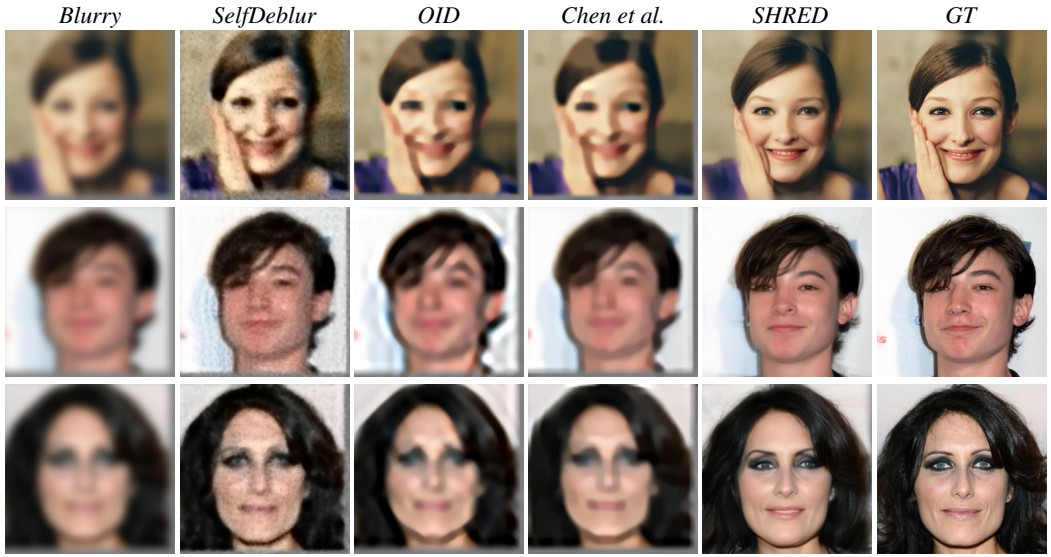

Figure 16: Visual results of blind deconvolution using a random Gaussian kernel with size $41 \times 41$ on CelebA.

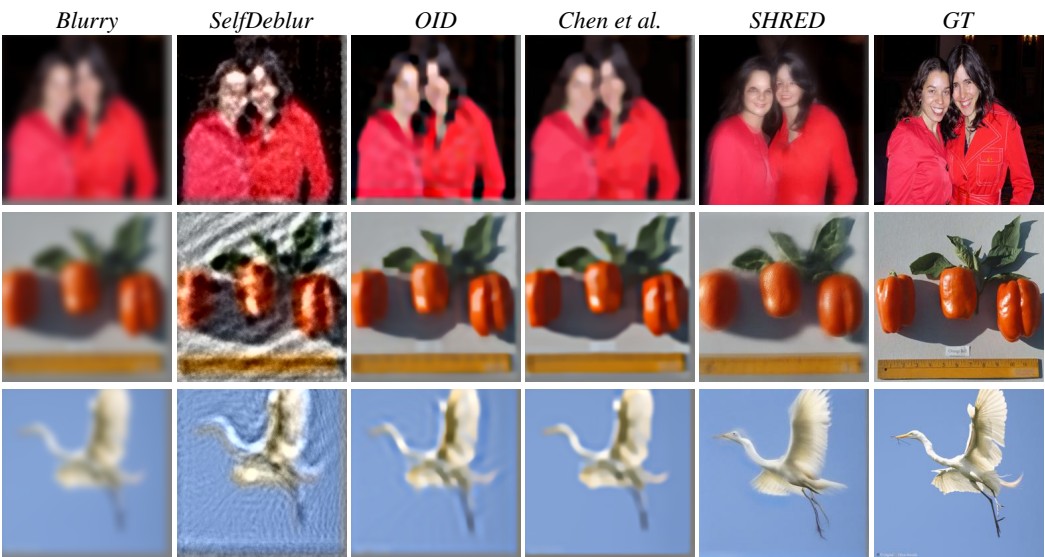

Figure 17: Visual results of blind deconvolution using a random Gaussian kernel with size $41 \times 41$ on ImageNet.

