# OpenReview forum: "Zero-shot Image Restoration via Diffusion Inversion"
_ICLR.cc/2024/Conference — Submitted to ICLR 2024_

### Official Review · Reviewer_9MEL · 2023-10-13

**Soundness:** 2 fair
**Presentation:** 2 fair
**Contribution:** 2 fair
**Rating:** 3
**Confidence:** 5

**Summary:**

This paper proposes a new method using diffusion models for zero-shot image restoration. Specifically, the authors use pre-trained DDIM networks as the prior, then optimize the latent, i.e., the initial noise $\mathbf{x}\_{T}$, to minimize the data consistency term to achieve image restoration. The proposed method achieves comparable results to state-of-the-art methods in several metrics such as LPIPS and FID.

**Strengths:**

A lot of work has been proposed using diffusion models for zero-shot image restoration. However, the idea of optimizing the inversion latent is new, and the authors verified that this type of method is feasible and can achieve plausible results.  The author gives the results on multiple tasks, evaluates the performance to a certain extent, and also provides some ablation analysis, which has a certain reference value for scholars in this field.

**Weaknesses:**

**Unclear motivation**: Similar to DPS [1], the proposed method (SHRED) also solves IR tasks in an optimization manner. The main difference is, DPS [1] optimizes the intermediate result $\mathbf{x}\_{t}$, but SHRED optimizes the initial noise $\mathbf{x}\_{T}$. I wonder what are the benefits of optimizing $\mathbf{x}\_{T}$ ? This article does not clearly point out SHRED's advantage, and it does not compare with DPS [1]. Besides, the authors claim that "This design choice does not alter the intermediate diffusion latents and thus provides more guarantees that the generated images lies in the in-distribution manifold", which lacks experimental or theoretical support. For example, the visualization of the optimized $\mathbf{x}\_{T}$ is necessary to judge whether it lies in the in-distribution manifold.
In summary, the necessity and superiority of this method are questionable.

**Insufficient experiments**: (1) No reports of PSNR. (2) Lack of comparison with optimization-based methods, e.g., DPS [1], GDP [2]. (3) The description of the experiment is not detailed enough, making it difficult for researchers to make effective evaluations based on the experimental results. For example, Table 3 does not tell the SR scale and the total steps for other methods. (4) Since SHRED seems to have a longer backpropagation chain than DPS [1], it may have a larger memory consumption. It would be better to compare the memory usage.

**Objectivity**: Lack of discussion of limitations.

**Typos**:  There are many typos in this article. The authors need to check carefully. For example, "the generated images lies in the in-distribution manifold" should use "lie" rather than "lies"; Errors in labeling the best and second best methods in Table 1 and Table 2.



References:

[1] Chung et al.,  Diffusion posterior sampling for general noisy inverse problems. ICLR 2023

[2] Fei et al., Generative diffusion prior for unified image restoration and enhancement. CVPR 2023

**Questions:**

Please see the weaknesses.

---

> ### Author Response · Authors · 2023-11-23
> **Rebuttal by Authors**
>
> We would like to thank the reviewer for the time and feedback.
>
> **Unclear motivation/ Lack of a theoretical support**:
>
> our guarantee that SHRED generates samples along the data manifold is **by construction**. SHRED works by sampling from the posterior distribution $P(x|y, \mu, \Sigma)$, where $\Omega_{real} = \\{x is real; \| H(x) - y\| = 0 \\}$  is its support. SHRED ensures realness (and so that the generated samples are in the data manifold) by directly using the diffusion reverse process as is (without modification).
>
> To give an idea, our method is equivalent to a brute force approach where we sample infinity initial noises $x_T$, apply the diffusion to get $x_0$ and at the end return the $x_T$, that minimizes $\| H(x_0) - y \|$. It is clear here that the generated $x_0$ are by construction in the data manifold. Instead of a computationally "impossible: brute force approach, we use gradient descent. So, the guarantee that the generated image by SHRED is within the data manifold is  **Equal** to the guarantee that when sampling a random noise and feeding through DDIM will generate a sample within the data manifold. Why? again, simply because the sampling path (between $x_T$ and $x_0$) of SHRED and  DDIM is exactly the same.
>
> **Insufficient experiments**: we will consider adding comparisons with the more recent works such as DPS[1] and GDP[2] in our next version of the paper.
>
> **Lack of discussion of the limitations**:  we thank the reviewer for pointing this out. We will take this into account in the next revision.
>
> **Typos**: we will improve the writing of the paper in its next version.

---

### Official Review · Reviewer_DyQ6 · 2023-10-20

**Soundness:** 1 poor
**Presentation:** 3 good
**Contribution:** 2 fair
**Rating:** 3
**Confidence:** 4

**Summary:**

Authors propose a technique for image restoration leveraging pre-trained diffusion models that can be used without any training data. The diffusion model acts as a prior for natural images. The method optimizes the initial noise realization in the reverse diffusion process only and thus does not rely on extra guidance terms such as DPS to enforce data consistency. Thus has the potential for improved image quality as the learned diffusion process is not perturbed by additional terms in the update. Experiments on large-scale image datasets demonstrate the performance of the proposed technique and comparison is made with other diffusion-based solvers.

**Strengths:**

- The core idea of optimizing over the noise input of the reverse process is interesting. Although it is conceptually very similar to replacing the convolutional network in Deep Image Prior with a diffusion model, the specific combination proposed in this paper is original to the best of my knowledge.
- Diffusion posterior sampling and similar methods suffer from the issue of perturbing the noisy data manifolds visited by the diffusion process thus leading to instabilities and potential poor performance on certain samples. Thus, techniques such as the one proposed constitute a valuable effort to tackle this problem.
- The paper is more or less clear and easy to follow, with some issues of equation formatting and typos.

**Weaknesses:**

- The proposed method is computationally very costly. The outer optimization needs to differentiate through several (n = 10+) chained calls of a large score model in each iteration, with an overall N (N=100+) outer loop iterations, per image. Thus, it requires at least 1000 NFEs plus the heavy cost of N backpropagations through a large, chained model.
- There are lots of hyperparameters that need to be tuned (step size, N, $\delta t$) and as the optimization needs to be solved on a sample-by-sample basis it is not clear how much variation in optimal hyperparameters can occur.
- I have serious doubts about the experiments:
    1) There is no comparison with DPS which is a well-known diffusion-based solver that has better performance than the included competing techniques and source code is made public. For instance, for 4x SR *with noise* DPS reports far better LPIPS than any techniques in this paper.

    2) Distortion metrics such as PSNR, NMSE and SSIM are completely missing, which are crucial in evaluating inverse problem solvers.

    3) A very small (100) number of samples is used for evaluation. In other competing methods it is standard to use a 1000-sample validation split. Thus the results are not necessarily reliable and it is very difficult to compare to existing published results (DDRM, DDNM, DPS). Furthermore, since FID heavily depends on the number of samples used in the generated distribution, the reposrted FIDs are not compatible with the ones reported in competing method's original papers.

    4) I have doubts about the reported timing results in Table 3. SHRED is reported as approx. 5x slower than DDRM. According to the DDRM paper, they use 20 NFEs. How is the reported timing possible, when SHRED uses 100 outer loop iterations with 10 NFEs in each outer loop (total 1000 NFEs) plus the additional cost of 100 backpropagation?

    5) The robustness experiments could be more rigorous. Instead of showing some good looking samples, it would be more meaningful to quantify the variation of image quality metrics for the validation dataset over 5-10 samples.

    6) The framework is developed for noisy inverse problems, however there are no experiments for the noisy case. Reconstruction performance under measurement noise is crucial in evaluating the utility of the algorithm.

**Questions:**

- What are the memory requirements of backpropagation through the loss, where the score is sequentially called several times to produce the loss? Is checkpointing used to make this possible?

- How does the method compare with DPS and other methods already presented in the paper (DDRM, DDNM) in terms of both perceptual and distortion metrics and on the standard 1000 samples from ImageNet and CelebA?

- How is the discrepancy in point 5) under Weaknesses explained with respect to timing?

- How does the method perform on noisy inverse problems?

- Why is the technique framed as a linear inverse problem solver? The linearity of the operator is not exploited.

---

> ### Author Response · Authors · 2023-11-23
> **Rebuttal by Authors**
>
> We would like to thank the reviewer for the time and feedback.
>
> **Doubts about the quantitative results:**
> we will consider adding more quantitative comparisons with recent works such as DPS using the same experimental setting (1000 samples). The reported results in our paper can't be directly compared with the ones published in the competing method's papers  as we don't use the same set/number of images.
>
> **No distortion metrics are used**:  The restoration tasks that we consider in our paper involve severe degradation (inpainting, x8, x16 super resolution), so the range of the plausible solutions is wide. The PSNR between the ground truth $x$ and reconstructed image ($PSNR(x, reconstructed)$) considers only a unique solution and advanges an over-smoothed result. For this, we followed [1] and [2] and report only perceptual metrics such as FID and LPIPS. However, we will consider adding a metric capturing the fidelity such as $PSNR(y, H(reconstructed))$ where $y$ is the degraded input image and H(reconstructed) is the degraded reconstructed image.
>
> **Doubts about the reported timing**: In our reported results for both DDNM and DDRM, we use number of denoising steps = 100 for the superresolution task. For SHRED, we use 50 optimization iterations and for every optimization iteration, we use 10 denoising steps ($\delta t = 100$). So at the end, the computational time for SHRED is approximately $\times 5$ times the one of DDNM and DDRM as shown in Table 3.
>
> **Quantitative metrics for the robustness experiments**: we will consider this point in our next revision.
>
> **Noisy inverse problems**: We report the result of SHRED for the case real-world image enhancement in Figure 7, where the images are noisy. Concerning the quantitative results, we will consider adding this case in the next version.
>
> **SHRED is only a linear inverse problem solver?**:  one of the advantage of SHRED compared to the other existing diffusion such as DDNM and DDRM is that can equally handle linear and non-linear inverse problems. In Figure 8 of our appendix, we show the result of (non-linear) non-uniform image deblurring. We will more point out this aspect in our next version of the paper.
>
> [1] Diffusion posterior sampling for general noisy inverse problems. ICLR 2023
>
> [2] Improving Diffusion Models for Inverse Problems using Manifold Constraints. Neurips 2022.

---

### Official Review · Reviewer_bZYN · 2023-11-05

**Soundness:** 3 good
**Presentation:** 3 good
**Contribution:** 2 fair
**Rating:** 5
**Confidence:** 2

**Summary:**

The paper presents SHRED (zero-SHot image REstoration via Diffusion inversion), a new image restoration method using pre-trained diffusion models. SHRED uniquely maintains the integrity of the data manifold by not altering the reverse sampling process during restoration. It optimizes the initial diffusion noise to reconstruct high-quality images efficiently, avoiding the need for model retraining or fine-tuning. SHRED demonstrates superior performance on various image restoration tasks, achieving state-of-the-art results in zero-shot benchmarks.

**Strengths:**

**Strengths of the Paper:**

1. **Originality:**
   - The introduction of SHRED represents a new direction in leveraging pre-trained diffusion models for image restoration tasks. The approach of not altering the reverse sampling process is a departure from previous methods, addressing limitations from prior results.

2. **Quality:**
   - The quality of the research is evident in the comprehensive experimental validation across various tasks such as inpainting, super-resolution, and blind deconvolution. The use of well-established metrics like FID and LPIPS lends credibility to the reported results.
   - The state-of-the-art performance of SHRED, as demonstrated through quantitative and qualitative evaluations, underscores the method's effectiveness.

3. **Clarity:**
   - Clarity is one of the paper's strengths, with a well-organized presentation of the content. The clarity in writing ensures that the concepts are accessible and the results are understandable.
   - The background provided on DDPM and DDIM is thorough, facilitating a clear understanding of the advancements SHRED brings to the field.

4. **Significance:**
   - The paper is significant in its potential applicability to a broad spectrum of image restoration tasks, demonstrating the adaptability of SHRED to different challenges without the need for retraining.

The paper's contributions are presented with clarity and are supported by solid empirical evidence, making it a valuable addition to the literature on diffusion models and image restoration.

**Weaknesses:**

**Weaknesses of the Paper:**

1. **Guarantee of Data Manifold Integrity:**
   - The paper positions SHRED as a method that maintains the integrity of the data manifold during image restoration, which is a central claim for its novelty. However, the paper lacks a rigorous demonstration or proof that the samples generated by SHRED indeed lie on the data manifold. This is a significant gap, as the main criticism of prior methods is their potential deviation from the manifold. To strengthen this claim, the authors could provide empirical evidence or a theoretical guarantee, possibly through visualizations of the manifold or quantitative measures that can assess this aspect.

2. **Comparison with MCG from Chung et al. (2022b):**
   - The paper does not provide a detailed comparison with the MCG method proposed by Chung et al. (2022b), which also aims to correct samples to ensure they are on the data manifold. A deeper theoretical and empirical analysis comparing SHRED with MCG would be beneficial. This could include side-by-side comparisons on the same tasks, using the same metrics, and a discussion on the theoretical underpinnings of both methods. Such a comparison would be valuable for readers to understand the relative merits and trade-offs of these approaches.

3. **Computational Efficiency:**
   - In Table 3, SHRED is slower than DDRM and DDPM, which could limit its practicality for real-world applications where computational resources or time are constrained. The authors could explore ways to improve the efficiency of SHRED, perhaps by optimizing the algorithm or by proposing a more computationally efficient variant that maintains most of the method's benefits.

4. **Novelty and Originality in Mathematical Derivation:**
   - The mathematical derivation of SHRED's methodology does not appear to be a novel contribution, which may lead to questions about the paper's originality. The authors could strengthen this aspect by clearly delineating the novel components of their mathematical approach, contrasting it with existing methods, and discussing how these novel aspects contribute to the method's performance.

**Questions:**

To address the above weaknesses, the authors could consider the following actions:

- Provide empirical evidence or theoretical justification for the claim that SHRED generates samples along the data manifold.
- Conduct a thorough comparison with MCG, including both theoretical and empirical analyses.
- Investigate and propose methods to improve the computational efficiency of SHRED.
- Clarify the novelty in the mathematical derivation of SHRED, differentiating it from existing approaches.

By addressing these points, the authors could significantly strengthen the paper and its contributions to the field.

---

> ### Author Response · Authors · 2023-11-22
> **Rebuttal by Authors**
>
> We would like to thank the reviewer for the detailed review, the constructive comments and suggestions.
>
> **Theoretical justification for the claim that SHRED generates samples along the data manifold**: our guarantee that SHRED generates samples along the data manifold is **by construction**. SHRED works by sampling from the posterior distribution $P(x|y, \mu, \Sigma)$, where $\Omega_{real} = \\{x is real; \| H(x) - y\| = 0 \\}$  is its support. SHRED ensures realness (and so that the generated samples are in the data manifold) by directly using the diffusion reverse process as is (without modification).
>
> To give an idea, our method is equivalent to a brute force approach where we sample infinity initial noises $x_T$, apply the diffusion to get $x_0$ and at the end return the $x_T$, that minimizes $\| H(x_0) - y \|$. It is clear here that the generated $x_0$ are by construction in the data manifold. Instead of a computationally "impossible" brute force approach, we use gradient descent. So, the guarantee that the generated image by SHRED is within the data manifold is  **Equal** to the guarantee that when sampling a random noise and feeding through DDIM will generate a sample within the data manifold. Why? again, simply because the sampling path (between $x_T$ and $x_0$) of SHRED and  DDIM is exactly the same.
>
> **Computational efficiency boost**: we control the efficiency of SHRED by the hyperparameter $\delta t$ which defines the jump length in DDIM sampling. We will consider thinking of other potential ways to further improve the efficiency.
>
> **Novelty in the mathematical derivation of SHRED**:
> we argue that SHRED is an original inverse problem solver. Our approach is orthogonal to all the existing diffusion based methods as being the first method based on diffusion inversion. Our method takes advantage of the bijection property between $x_T$ and $x_0$ in DDIM and  proposes an efficient optimization scheme (with respect only to the initial noise $x_T$) for DDIM inversion for solving inverse problems through playing on its hyper-parameters such as $\delta t$. although being simple, our inversion scheme is original as it is the first optimization-based one. The existing method for inverting DDIM is done by solving the reverse ODE. In our case, reversing the ODE will not work as the degraded image is out of distribution and so we propose an optimization based approach. A big part of  SHRED derivation is based on the exact DDIM derivation as we are inverting DDIM at the end.

---

### Official Review · Reviewer_Rmtd · 2023-11-08

**Soundness:** 1 poor
**Presentation:** 1 poor
**Contribution:** 2 fair
**Rating:** 3
**Confidence:** 5

**Summary:**

This paper proposes a new method called SHRED (zero-SHot image REstoration via Diffusion inversion) for solving image restoration problems using a pre-trained diffusion model. Current diffusion model-based methods for image restoration modify the reverse sampling process to satisfy consistency with the corrupted input image. However, this can cause the generated image to deviate from the true data distribution (According to the authors). The proposed SHRED avoids this issue by casting image restoration as an optimization problem over just the initial noise vector that is input to the diffusion model. To make this computationally feasible, SHRED utilizes the ability of DDIM to skip ahead in the diffusion process with large timesteps. This allows efficient inversion of the diffusion model for optimization. SHRED is evaluated on image inpainting, super-resolution, compressed sensing, and blind deconvolution tasks.

**Strengths:**

The method described is rather simple and intuitive. We need optimization-based diffusion inversion technique.

**Weaknesses:**

Here are my concerns regarding this paper:

Firstly, the writing of the paper requires substantial improvement. The method described in the paper is not complex; it essentially pertains to an optimization-based inversion method supplemented with some implementation tricks. However, the paper is very difficult to understand. It requires multiple readings to identify the important optimization objectives and to guess the optimization methods. Figure 1 is also difficult to comprehend. Section 3.1 seems unnecessary. The authors need to carefully revise the structure of the paper to improve the efficiency of information delivery. The current version is not suitable for publication.

Secondly, there is a large body of literature related to GAN inversion that has not been discussed. Many existing works are actually very relevant to the methods of this paper, but they have not been carefully considered.

Lastly, the experiments presented in the paper are somewhat insufficient. There is no quantitative data supporting the discussion on the blind problems. Moreover, the paper's inversion method seems to not address the prompt issues of the diffusion model.

Overall, the method described is rather simple and intuitive. My rating is not based on the method. There needs to be a description of an optimization-based diffusion inversion technique. However, the manuscript is not adequately prepared at this stage. This is the main reason for my negative review.

**Questions:**

About the method and Figure 1:

$x_T$ is randomly sampled from a Gaussian distribution. Why $x_{0|T}$ can be such am image with similar face with $y$? there is no $y$ involved in this process. I can guess what is actually done in this process. But from the paper, it just don't make sense. I partly consider this as the problem in writing.

Can the author provide any other supps to show their actual method, such as code?

---

> ### Author Response · Authors · 2023-11-22
> **Rebuttal by Authors**
>
> We would like to thank the reviewer for the time and feedback.
>
> **Better writing/presentation**: we thank the reviewer for pointing this out. We will improve the paper presentation in our next revision.
>
> **GAN inversion discussion**: we discussed some of the representative works of the GAN inversion based methods (both the optimized-based and learning based ) in "Methods based on GAN inversion" section of the Related work. Due to space limitation and that those methods are no longer sota, we didn't give more space to this discussion. However, We will consider enlarging it in our next revision.
>
> **More experiments for the blind case**: One of the restoration tasks that we consider is blind deconvolution. The corresponding results are shown in Table 1 and Table 2. We will consider showing the results of more blind restoration tasks in the next revision.
>
> However, we don't understand what the reviewer means by  "the paper's inversion method seems to not address the prompt issues of the diffusion model"
>
> **Code**: we included our code in the supplementary material. At the first iteration of the optimization $x_T$ is a random noise and $x_0$ is not similar to $y$, throughout the gradient descent steps using the loss $\| H(x_0) - y \|$, $x_0$ becomes more and more similar to $y$.

---

### Author Response · Authors · 2023-11-23
**Rebuttal by Authors**

We would like to thank all the reviewers for their time and feedback. We attached our code in the supplementary material.

---

### Meta-Review · Area_Chair_tc5U · 2023-12-11

**Metareview:**

This paper proposes a zero-shot method, SHRED, to solve image restoration problems by using pre-trained diffusion models. It formulates the the inverse restoration problem as a latent estimation problem by exploiting the deterministic correspondence between noise and images in DDIM. The proposed method is simple and easy to understand.

However, all reviewers are not in favor of this paper due to expensive computational costs, poor writings, limited experimental evaluations, and so on.

Based on the recommendations of reviewers, the paper is not ready for ICLR. The authors are encourage to revise the paper accordingly.

**Justification For Why Not Higher Score:**

N/A

**Justification For Why Not Lower Score:**

N/A

---

### Decision · Program_Chairs · 2024-01-16

Reject